# Hereditary Spastic Paraplegia Type 11—Clinical, Genetic and Neuroimaging Characteristics

**DOI:** 10.3390/ijms242417530

**Published:** 2023-12-15

**Authors:** Justyna Chojdak-Łukasiewicz, Katarzyna Sulima, Anna Zimny, Marta Waliszewska-Prosół, Sławomir Budrewicz

**Affiliations:** 1Department of Neurology, Wroclaw Medical University, 50-556 Wroclaw, Poland; justyna.chojdak-lukasiewicz@umw.edu.pl (J.C.-Ł.); katarzyna.sulima91@gmail.com (K.S.); slawomir.budrewicz@umw.edu.pl (S.B.); 2Department of General Radiology, Interventional Radiology and Neuroradiology, Wroclaw Medical University, 50-556 Wroclaw, Poland; anna.zimny@umw.edu.pl

**Keywords:** spastic paraplegia, autosomal recessive HSP, spatacsin, ears of the lynx, MRI

## Abstract

Hereditary spastic paraplegia (HSP) is a heterogeneous group of genetically determined diseases, characterised by progressive spastic paraparesis of the lower limbs, associated with degeneration of the corticospinal tract and the posterior column of the spinal cord. HSP occurs worldwide and the estimated prevalence is about 1–10/100,000, depending on the geographic localisation. More than 70 genes responsible for HSP have been identified to date, and reports of new potentially pathogenic variants appear regularly. All possible patterns of inheritance (autosomal dominant, autosomal recessive, X-linked and mitochondrial) have been described in families of HSP patients. Among the autosomal recessive forms of HSP (AR-HSP), hereditary spastic paraplegia type 11 is the most common one. We present a patient with diagnosed HSP 11, with a typical clinical picture and characteristic features in additional diagnostic tests.

## 1. Introduction

Hereditary spastic paraplegia (HSP) is a heterogeneous group of neurodegenerative conditions, characterised by progressive degeneration of the corticospinal tract and the posterior column of the spinal cord [1]. The clinical picture consists primarily of progressive weakness and spasticity of the lower limbs, but some patients may present with other symptoms [1,2]. The primary symptom of HSP in most cases is difficulty walking due to muscle weakness and spasticity in the legs (varying from minimal to severe). The symptoms coexist with other neurologic symptoms. The diagnosis of HSP is difficult because HSP is characterised by wide genetic heterogeneity, with autosomal dominant, autosomal recessive, X-linked and mitochondrial patterns of inheritance [2,3]. Moreover, sometimes, there are differences in the nature and severity of symptoms between individuals who have exactly the same genetic type of HSP.

More than 100 loci or genes associated with HSP have been described [3,4,5]. HSP may be classified either as “pure” when it comes mainly with spastic paraparesis or as “complex” when additional symptoms are found [6].

Hereditary spastic paraplegia type 11 (HSP 11) is the most common subtype of autosomal recessive HSP and is the major cause of HSP with the thin corpus callosum (HSP-TCC) [2,4]. The typical initial signs are gait disturbances followed by cognitive impairment and other neurological or systemic manifestations. The clinical feature of HSP type 11 is variable and the insidious symptoms can lead to delays in diagnosis [4,5,6].

We describe a case of a young women with HSP type 11 to emphasise the role of the diagnosis based on clinical picture, as well as some specific MRI signs, in the diagnosis of recessive forms of HSP. This broad clinical spectrum and the absence of positive background family history make the recessive forms of HSP a difficult group to diagnose. We would also like to highlight the growing benefits offered to the clinician by the use of modern genetic testing techniques.

## 2. Clinical Report

A 21-year-old woman, with no prior comorbidities, presented with a 3-year history of progressive lower limb weakness, accompanied by stiffness and painful “cramps”. Moreover, during this period, the patient began to notice a deterioration in memory and attention, resulting in poorer learning outcomes and not completing her high school education. The patient’s early psychomotor development was normal; she achieved all the milestones during childhood. The family history was unremarkable. Initially, at the age of 19, the patient was hospitalised in another neurology centre, where neuroimaging and electrophysiological studies did not reveal any abnormalities; the cerebrospinal fluid was normal, and the presence of oligoclonal bands and anti-Borrelia burgdorferi antibodies were excluded. Hereditary spastic paraplegia (HSP) was suggested, and the patient was referred to a genetic counsellor.

Genetic analysis by whole-exome sequencing with the next-generation sequencing (NGS) method was performed, indicating compound heterozygous variations in the patient’s SPG11 gene: the c.4307_4308del p. (Gln143ArgfsTer7) variant on the paternal allele and the c733_734del p (Met245ValfsTer2) variant on the maternal allele. Both variations were previously reported in the literature as pathogenic for autosomal recessive HSP.

Neurologic examination revealed severe spasticity and moderate muscle weakness of the lower limbs, with the slight predominance of the right side, together with other pyramidal signs, such as hyperreflexia, bilateral ankle clonus and Babinski sign. The upper limbs were spared. The patient presented typical spastic gait, with a tendency to rotate feet inwards. No muscle atrophy was observed. Cerebellar function was normal. The patient did not report any urinary urgency.

Magnetic resonance imaging (MRI) of the brain showed T2-weighted/fluid-attenuated inversion recovery (FLAIR) periventricular white matter hyperintensities at the frontal horns of the lateral ventricles, so called “ears of the lynx” sign. Similar lesions, although less pronounced, could be seen in the neighbourhood of the posterior horns of lateral ventricles. Moreover, the corpus callosum was visibly thinner in its rostrum, genu and trunk (Figure 1). The cervical and thoracic spine MRI presented no abnormalities. Interestingly, the control nerve conduction study revealed sensorimotor polyneuropathy, with predominantly demyelinating features. The electroencephalogram (EEG) was normal. Visual evoked potentials (VEP) were recorded, showing no abnormalities. Also, the ophthalmic exam excluded any pathologies in the eyes. There were no significant changes in the laboratory tests; the serum ceruloplasmin level was within the normal range. Neuropsychological assessment showed problems with concentration and short-term memory.

The project was approved by the Commission of Bioethics at Wroclaw Medical University (number of permission: KB-313/2013).

## 3. Discussion

Hereditary spastic paraplegia type 11 is the most common autosomal recessive type of HSP, and accounts for up to 8% of all cases. The global prevalence rate of HSP 11 is estimated at 0.34 per 100,000 [7]. The prevalence is higher in the Mediterranean region and in the Middle East, especially in populations with considerable consanguinity [2,8,9]. The age of onset is variable and varies from 4 to 36 years (mean age is 14.3 years). Later onset, between 50–60 years, is less frequent [10,11].

The cardinal symptoms of HSP 11 include slowly progressive spastic paraparesis with sphincter disturbances [9]. Additional clinical features are intellectual disability with learning difficulties in childhood and/or progressive cognitive retardation, peripheral neuropathy (axonal, motor or sensorimotor), cerebellar signs, parkinsonism and pseudobulbar involvement (Table 1) [2,9,11]. Other signs, such as obesity or peripheral lymphoedema, have also been reported. Cognitive dysfunction is present in 80–100% of patients with HSP 11 [11,12]. It can be manifested by either mental retardation or mental decline. According to the results of neuropsychological assessment, executive function, attention and visual perception are especially disturbed [13]. The phenotype of the disease is complex and varies among family members [12,13,14].

Hereditary spastic paraplegia is characterised by high genetic heterogeneity, as all known forms of inheritance (autosomal dominant, autosomal recessive, X-linked and mitochondrial) are being reported in the literature [8,14,15,16,17]. Also, 13–40% of cases are sporadic, which means there is no family history [8]. So far, more than 70 loci have been described, some of them only in single families. HSP 11 is caused by pathogenic variants in the SPG11/KIAA1840 gene, mapping to chromosome 15p, pp. 13–21 [16,18,19]. To date, there have been more than 180 SPG11 pathogenic variants identified, and new ones are still emerging. All types of variants have been described, and among them, frameshift variants, such as small deletions or insertions, are the most frequent ones (54%), followed by nonsense (23.20%) and splice site variants (19.33%) [9].

According to Kawari et al., missense and splice site variants tend to be associated with later onset of the disease, while—as reported by Kara et al.—they might cause milder symptoms [10,20]. Since SPG11 is a large gene, including 40 exons, large deletions or duplications are quite often detected, accounting presently for about 10% of all variants, as reported by Du J [9]. There is no evident genotype–phenotype correlation in HSP 11, as the same variant in one family may lead to different clinical presentation [21]. This observation indicates that the full phenotype spectrum may be conditioned by other genetic or environmental factors [22]. It is noteworthy that SPG11 pathogenic variants have been reported in association with autosomal recessive Charcot–Marie–Tooth (CMT) disease and slowly progressive juvenile-onset autosomal recessive amyotrophic lateral sclerosis (ALS) [18,21]. This might suggest that SPG11 correlates with some spectrum of clinical manifestation, resulting from upper and/or lower motor neuron degeneration. The SPG11/KIAA1840 gene encodes spatacsin, a 2443-amino acid protein whose function is still under discussion. With a few exceptions, the vast majority of SPG11 pathogenic variants cause an early spatacsin truncation, leading to its loss of function. Based on the mice model, it has been postulated that the loss of spastactin function leads to the lipids accumulation in lysosomes by altering their clearance process. There is a hypothesis that spatacsin acts as a carrier protein, as it probably contains some transmembrane regions, or, alternatively, may play a role in sorting lipids within lysosomes. Either way, the precise mechanism of its function needs further investigation [23,24]. The SPG11 pathogenic variants are scattered throughout the entire gene and may be located in almost all of the 40 exons; in simple terms, there is no mutational hotspot. Hence, whole-gene analysis is needed in clinical practice. There are a couple of approaches available for clinicians to test HSP genetically, which can be used separately or in combination. When the clinical presentation is suggestive for type 11 HSP, then gene-targeted testing can be performed. However, if the phenotype cannot be distinguished from other subtypes of the disease, then comprehensive genomic testing, such as exome or genome sequencing, should be implemented.

One of the most practical methods, which happened to be effective also in the presented case, is next-generation sequencing (NGS), as it allows for the screening of all exons. Nonetheless, it has its limitations, as it does not detect whole-gene deletions or duplications [2,25]. If the NGS method is not diagnostic, an exome array should be considered. Despite the availability of all these molecular methods, a genetic-based diagnosis is not made in 51–71% of all patients with suspected HSP [8]. Spinal cord imaging serves an important role in determining the differential diagnosis; hence, it should be performed in every patient with suspected hereditary spastic paraplegia. However, its role in phenotypic characterisation is rather minor [26]. Bearing in mind the primary HSP pathophysiologic mechanism, which is the long-projecting axon degeneration, it is not surprising that spinal cord volumetric reduction is present in most patients. Such atrophy, however, might not always be visualised by routine MRI, especially—like in the presented case—at the beginning of the disease. Moreover, it lacks specificity and can be found in other neurodegenerative conditions. With regard to HSP type 11, there are morphometric studies showing more severe spinal cord atrophy in patients with longer disease duration. In terms of brain MRI, the situation is more complex due to the variety of potential lesions, among which only a few are characteristic. The hallmark of SPG11 is the thinning of the corpus callosum [27]. However, this sign is not unique, and might be present in about one-third of recessive HSP cases—this group of disorders is known as hereditary spastic paraplegias with this corpus callosum (HSP-TCC). According to a genetic assessment of an Italian cohort of HSP-TCC patients (n = 61), SPG 11 accounted for 26.2%, and thereby was the most frequent type [26]. However, there are also rare cases of SPG 11 with the absence of corpus callosum involvement [28]. There is uncertainty as to whether the selective damage to the corpus callosum is rather due to atrophy or hypoplasia. França, Marcondes C Jr et al. described two patients with a progressive volumetric reduction in the corpus callosum over a period of one year, which stands in accordance with our case [29].

Another interesting neuroimaging finding, probably even more distinctive, is the so-called “ears of the lynx” sign. This term refers to an abnormal signal intensity of the forceps minor of the corpus callosum (genu fibres), which presents as hypointensity in T1-weighted images and hyperintensity in T2-FLAIR-weighted images. When looking at axial scans, the sign resembles a lynx’s ears, which it owes its name to. One more form of autosomal recessive HSP where this abnormality might be found, yet rarely, is SPG 15. Consequently, the “ears of the lynx” sign is not pathognomonic for SPG11, but when present together with the thin corpus callosum, it might increase the sensitivity and specificity of the visual evaluation of MRI images. The issue of whether this sign is present before the symptoms’ onset is unclear and requires further investigation. Clinicians should be aware of potential mimics, such as gliosis at the calloso-caudate angle of the frontal horn of the lateral ventricles, observed in some healthy individuals, as well as multiple sclerosis.

The broad clinical variability of HSP type 11 suggests that neuronal damage extends far beyond corticospinal tracts. Researchers use advanced neuroimaging tools to investigate the actual distribution of cerebral changes and its relationship to clinical features [30]. A study on patients with SPG11, using voxel-based morphometry (VBM), led to the identification of a significant grey matter (GM) volumetric reduction in the thalamus and basal ganglia, as well as small portions of cortex, restricted to the precentral and paracentral gyri. These structures are part of significant motor and cognitive networks; particularly, the damage to the thalamus, together with the corpus callosum atrophy, might be the cause of dementia and behavioural changes. The authors of the same study revealed notable and diffuse white matter (WM) abnormalities. VBM enabled the confirmation of severe corpus callosum atrophy, while diffusion tensor imaging (DTI) identified microstructural lesions to the cerebellum, corpus callosum, cingulate gyrus and subcortical white matter of the temporal and occipital lobes. Interestingly, there is a positive correlation between GM damage and the disease duration, which has not yet been proven for WM [26]. There are conflicting reports of whether WM abnormalities correspond to the disease severity [26,30].

The long-term prognosis and life expectancy of hereditary spastic paraplegia 11 is limited. Currently, there is no causal treatment for HSP patients. The therapeutic approach should focus on symptomatic treatment and physical and speech therapy. The first line of treatment includes antispastic drugs (i.e., baclofen, tizanidine, dalfampridine) and orthopaedic solutions. Therapeutic options can include botulinum toxin therapy and surgical implantation of the baclofen pump. Chen et al. report that the application of high-frequency repetitive transcranial magnetic stimulation (HF-rTMS) to the bilateral leg area of M1 (M1-LEG) is beneficial for SPG11-HSP [24]. At present, several clinical trials are conducted in HSP 11.

Miglustat, which inhibits and inhibits glycosphingolipid synthesis, reduces the accumulation of GSL both in neurons from Spg11-mutated mice and in fibroblast-derived cells of individuals with SPG11 [31]. The complicated pathomechanisms make it difficult to find universal treatment for the disease [32].

## 4. Conclusions

Hereditary spastic paraplegia is a group of clinically and genetically diverse diseases, in which the main feature is a slow, progressive weakness of the lower limbs. The onset of signs may occur in child- or adulthood. Furthermore, the severity and progression of symptoms will vary from person to person. Various types of HSP have been reported, without a clear genotype–phenotype correlation, so there are many conditions that mimic HSP.

We describe a patient with diagnosed and genetically confirmed hereditary spastic paraplegia type 11, with typical clinical symptoms and characteristic MRI findings, such as “ears of the lynx” sign and thinning of the corpus callosum. Genetic examination by the whole-exome sequencing method (WES) is a useful tool in the diagnosis of rare multi-gene diseases, as in the case of hereditary spastic paraplegia. The diagnosis of HSP should be considered in patients with spastic paraplegia, especially in case of an atypical clinical course and/or symptomatology.

## Figures and Tables

**Figure 1 ijms-24-17530-f001:**
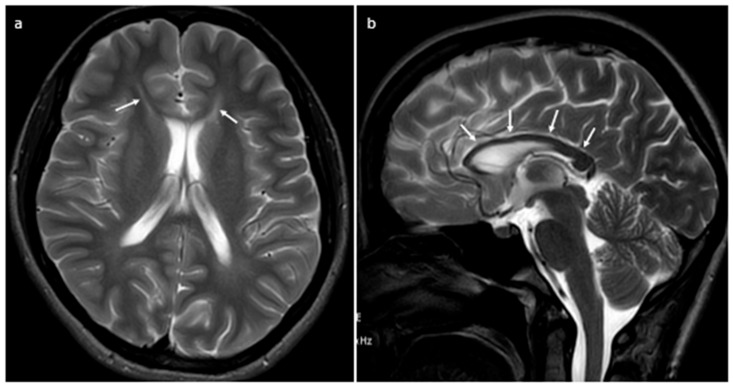
Magnetic resonance T2-weighted images in axial (**a**) and sagittal (**b**) planes. Axial image shows high T2 signal intensity at the tips of the frontal horns of the lateral ventricles called “ears of the lynx” sign (arrows), while sagittal image clearly demonstrates hypotrophy of the corpus callosum (arrows).

**Table 1 ijms-24-17530-t001:** Clinical features of hereditary paraplegia type 11 [2,5,9,11].

Typical Symptoms	Less Frequent Symptoms
progressive weakness and spasticity of the lower limbs [2] mild intellectual disability with learning difficulties in childhood and/or progressive cognitive decline [2,9] peripheral neuropathy [5] dysarthria and/or dysphagia [2,5] hyperreflexia in the upper limbs [9]	cerebellar signs [2,9] retinal degeneration (Kjellin syndrome) [9,11] pes cavus, scoliosis [2,11] extrapyramidal symptoms [9]

## Data Availability

Data are contained within the article.

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
