# Peer review of "Hereditary Spastic Paraplegia Type 11—Clinical, Genetic and Neuroimaging Characteristics"

_ijms, 2023, doi:10.3390/ijms242417530_

Round 1
Reviewer 1 Report
Comments and Suggestions for Authors
In this case report, the authors described a case of the hereditary spastic paraplegia type-11 of 21-years old female patient.
The idea of this study - is interesting; but his manuscript needs some improvements and corrections before publishing may be possible.
General points:
Please add a list of abbreviations before References section to your manuscript.
Special points:
Introduction
Please describe more information known up to date about this disease.
Please justify more exactly the importance of your case report for physician.
Please describe more exactly the difficulty of the diagnostic of this disease.
Lines 23-29: please add ore references at the end of each these sentences.
Discussion
Lines 78-90: please add ore references at the end of each these sentences.
Lines 99-100: please add multiple references at the end of this sentence.
Please add more information about the treatment of the HSP.
Table 1: please add an appropriate reference for each symptom described by you in this table.
Materials and Methods
First of all, please add to this section the exactly name of the organization, the exactly date and the number of the permission of all your experiments.
Conclusion
Please add more conclusions and Future Perspectives section to your manuscript.
Please add some concrete suggestions for neurologists and another physician, how to manage the HSP.
Author Response
Dear Editor,
Dear Referees,
First of all we would like to thank the Editors and Reviewers for the time taken in reviewing our study. We highly value all the comments and truly believe that implementing these changes into our paper will improve the final manuscript.
On behalf of the authors,
Marta Waliszewska-Prosół
Thank you very much for your very valuable comments, we greatly appreciate the time you took to review our manuscript.
All suggestions and linguistic oversights have been corrected and included in the new version of the manuscript. Changes are highlighted in yellow in the new version and refer to both omitted citations and expansion of suggested parts of the text.
Reviewer 2 Report
Comments and Suggestions for Authors
A clear and concise description of a case report considering a patient with autosomal recessive HSP11. The clinical examination is thorough and strong case for the use of exome sequencing in diagnosis is made. An interesting consideration which is not explored is the carrier status and potential phenotypic effects of carrier status in the proband's parents, in this case and others with a similar inheritance pattern. (How were the parental contributions determined in this case)?
Comments on the Quality of English LanguageA couple of minor errors in English but insufficient for major changes, l23 the use the is redundant, l71 should be short-term memory, l118 amino acid
Author Response
Dear Editor,
Dear Referees,
First of all we would like to thank the Editors and Reviewers for the time taken in reviewing our study. We highly value all the comments and truly believe that implementing these changes into our paper will improve the final manuscript.
On behalf of the authors,
Marta Waliszewska-Prosół
Thank you very much for your very valuable comments, we greatly appreciate the time you took to review our manuscript.
All suggestions and linguistic oversights have been corrected and included in the new version of the manuscript.
Round 2
Reviewer 1 Report
Comments and Suggestions for Authors
Dear authors,
thank you for your corrections. Unfortunately, the authors do not addressed all my proposals.
Please answer all my proposals in your response letter point-by-point.
Once again:
Introduction
Please describe more information known up to date about this disease.
Please justify more exactly the importance of your case report for physician.
Please describe more exactly the difficulty of the diagnostic of this disease.
Table 1: please add an appropriate reference for each symptom described by you in this table.
Materials and Methods
First of all, please add to this section the exactly the exactly date and the number of the permission of all your experiments.
Author Response
Dear Editor,
Dear Referees,
First of all we would like to thank for the time taken in reviewing our study. We highly value all the comments and truly believe that implementing these changes into our paper will improve the final manuscript. Below you will find a point-by-point response.
On behalf of the authors,
Marta Waliszewska-Prosół
Reviewer 1
1. Introduction:
Please describe more information known up to date about this disease.
Please justify more exactly the importance of your case report for physician.
Please describe more exactly the difficulty of the diagnostic of this disease.
Answer: We have expanded the introduction section with additional information. At the same time, we would like to note that HSP is presented extensively in the discussion where we refer to individual clinical, genetic and radiological aspects. We did not want to repeat the same information twice.
2. Table 1: please add an appropriate reference for each symptom described by you in this table.
Answer: It has been completed.
3. Materials and Methods
First of all, please add to this section the exactly the exactly date and the number of the permission of all your experiments.
Answer: It has been completed: The project was approved by the Commission of Bioethics at the Wroclaw Medical University (number of permission: KB-313/2013).